# Transport Mechanism of Paracetamol (Acetaminophen) in Polyurethane Nanocomposite Hydrogel Patches—Cloisite^®^ 30B Influence on the Drug Release and Swelling Processes

**DOI:** 10.3390/ma17010040

**Published:** 2023-12-21

**Authors:** Justyna Strankowska, Małgorzata Grzywińska, Ewelina Łęgowska, Marek Józefowicz, Michał Strankowski

**Affiliations:** 1Institute of Experimental Physics, Faculty of Mathematics, Physics and Informatics, University of Gdańsk, Wita Stwosza 57, 80-308 Gdańsk, Poland; marek.jozefowicz@ug.edu.pl; 2Neuroinformatics and Artificial Intelligence Lab, Department of Neurophysiology, Neuropsychology and Neuroinformatics, Medical University of Gdańsk, Tuwima 15, 80-210 Gdańsk, Poland; 3Academia Copernicana Interdisciplinary Doctoral School, Nicolaus Copernicus University, Lwowska 1, 87-100 Toruń, Poland; ewelina.legowska@gmail.com; 4Department of Polymer Technology, Chemical Faculty, Gdańsk University of Technology, G. Narutowicza 11/12, 80-233 Gdańsk, Poland

**Keywords:** transport mechanism, diffusion, drug delivery, diffusion coefficients, swelling/release exponent, hydrogel, crosslinking, nanocomposite, paracetamol (acetaminophen), Cloisite^®^ 30B, clay

## Abstract

This article describes the swelling and release mechanisms of paracetamol in polyurethane nanocomposite hydrogels containing Cloisite^®^ 30B (organically modified montmorillonite). The transport mechanism, swelling and release processes of the active substance in nanocomposite matrix were studied using gravimetric and UV-Vis spectroscopic methods. Swelling and release processes depend on the amount of clay nanoparticles in these systems and the degree of crosslinking of PU/PEG/Cloisite^®^ 30B hydrogel nanocomposites. The presence of clay causes, on the one hand, a reduction in free volumes in the polymer matrices, making the swelling process less effective; on the other hand, the high swelling and self-aggregation behavior of Cloisite^®^ 30B and the interactions of paracetamol both with it and with the matrix, cause a change in the transport mechanism from anomalous diffusion to Fickian-like diffusion. A more insightful interpretation of the swelling and release profiles of the active substance was proposed, taking into account the “double swelling” process, barrier effect, and aggregation of clay. It was also proven that in the case of modification of polymer matrices with nanoparticles, the appropriate selection of their concentration is crucial, due to the potential possibility of controlling the swelling and release processes in drug delivery patches.

## 1. Introduction

Hydrogels are three-dimensional crosslinked structures composed of polymer chains linked together in a number of crosslinks. The desired properties of hydrogels, i.e., the ability to absorb water, nontoxicity, biodegradability, porosity, flexibility, good mechanical strength, mainly depend on the type of polymers and their chemical compositions, as well as on type and amount of crosslinking agent in the matrix. Temperature, pH, mechanical forces, solvents affect the stability of hydrogels; exceeding certain critical values of these parameters leads to matrix degradation [1,2]. Due to their unique properties, hydrogels have found many applications in tissue engineering, scaffolds, biosensors, and drug carriers [1,2,3,4,5,6]. Hydrogel drug delivery systems have become an interesting research area due to the possibility of designing systems with desired swelling and drug release properties.

Polyurethane nanocomposite hydrogels, containing sodium montmorillonite clay mineral, are a new class of hybrid materials that exhibit better mechanical properties, swelling abilities, and thermal stability than pure polymers, as presented in our previous papers [7,8,9]. The type of nanocomposite was determined based on XRD analysis. All the nanocomposite samples containing 0.5 and 1% of Closite^®^30B nanofiller are exfoliated, but for systems with 1% of Clo, a diffraction peak of very low amplitude is observed and is related to the presence of the nanofiller in the form of aggregated stacks in the polymer matrix. We do not observe a shift in this peak towards lower values of the 2θ angle; therefore, it does not form an intercalated structure. The mechanical properties of polyurethane (PU) materials were studied in the context of dynamic thermomechanical studies, where the storage modulus (E’), which is one of the indicators describing the performance of polyurethane nanocomposites, was analyzed. The developed systems, based on DMA studies, showed a higher E’ modulus for nanocomposites containing Cloisite^®^ 30B, both in the glassy and viscoelastic region in comparison to the nonmodified PU. The mechanical properties of nanocomposites were investigated using the DMA technique. We observed an improvement in mechanical properties by incorporation of clay and that can be controlled by varying the nanofiller content in these systems as well as molecular dynamics (measured using the NMR technique) of the polymer chains in PU/PEO nanocomposites depending on montmorillonite content in polymeric materials [7]. The size of the pores in the PU/PEG 4000 hydrogels was examined using the DSC technique—thermoporometry—and was presented in [8]. For the pure nanocomposite, the pore size was 4.27 nm, for 0.5% 4.57 nm, and for 1% Clo 6.84 nm.

Polyurethane materials are widely used as wound dressing, adhesives, and elastomers in skin tissue engineering, scaffolds, and drug delivery [10,11,12]. They represent a wide range of polymers that can be easily modified by changing reactive substrates. Thanks to their good mechanical and thermal properties and characteristic viscoelastic properties, they can be successfully used as dressing materials. They are some of the most biocompatible and safe for biological systems materials available today. Thanks to the ability to control and modify the structure of polyurethanes, it is possible to design systems with specific properties that may influence drug delivery processes, e.g., absorption, diffusion, solubility, erosion, and degradation, which makes it possible to use polyurethanes in a wide range of drug delivery systems [5]. Currently, mainly acrylates, polyurethanes, and silicones are used to produce patches for pharmaceutical use; unmodified silicones and acrylates have poor swelling properties, while polyurethanes can be chemically modified, e.g., crosslinked, to increase their sorption, which plays a key role in active substance release processes.

Clays are commonly used in pharmaceutical technology to control the release process, which depends on filler–polymer interaction, drug surface charge, network density, as well as nanocomposite structure. Montmorillonite is one of the most known clays in nanocomposite technology and is used in laboratories and industry because of its good compatibility with polymers [13,14]. It has been proven that the layered structure of montmorillonite affects the drug release process. Drug swelling and release behavior depend on clay content. Slower release and decrease in swelling properties were observed for different drugs (paracetamol, theophylline, xanthine) in chitosan/Cloisite 15A hydrogels with increasing clay content (5–9%), which was explained by the decrease in pore size of the matrix [15]. On the other hand, for relatively low montmorillonite contents (1–5%), better swelling properties were observed compared with nonfiller samples or high filler contents (10–25%) [16]. The reduction in swelling was also observed in CS-gpoly(AA-co-AAm)/PVP/MMT samples and explained as a consequence of the difference in osmotic pressure between the swelling medium and hydrogel samples [17].

In our previous paper, we also observed that for Cloisite^®^ 30B—a synthetic montmorillonite—the presence of clay nanoparticles hinders drug diffusion from the polyurethane matrix and slows down drug release. High content of nanofiller resulted in a decrease in swelling properties, as a consequence of the barrier and nanosize effects [18]. The release and swelling processes can therefore be controlled by the concentration of montmorillonite but also by the presence of hydrophilic soft segments such as polyethylene oxide (PEO) in polyurethane (PU) structures, which can result in increased size as the matrix swells, increasing drug molecule distance to diffuse, which in turn reduces the release rate [5].

The water absorption in hydrogels depends on network parameters, type of solutions, porosity, and crosslinking density. Typical swelling kinetics of the hydrogel consists of three stages: softening of dry hydrogel (very quick and difficult to observe), a very rapid diffusion of water to the hydrogel, relaxation of polymer chains until swelling equilibrium is reached. Swelling kinetics depends mainly on polymer crosslinking density; diffusion is controlled within a shorter time range followed by a relaxation process. Increasing crosslinking density causes a shortening in the duration of the diffusion process and accelerates it. Equilibrium hydrogel swelling is achieved as a balance of elastic forces that originate from the crosslinking network and swelling forces dependent on polymer dissolution, electrostatic interactions, and osmosis [1,2,7,19].

Paracetamol (N-acetyl-p-aminophenol) is one of the most widely used nonsteroidal, antipyretic, analgesic, and anti-inflammatory drugs. Dosage forms of paracetamol include tablets, suppositories, and suspension, and are intended for adults, children, and infants. Apart from the potential toxicity, oral intake of paracetamol causes damage to the liver, hepatic necrosis, loss of appetite, jaundice, stomach upset, thrombocytopenia, and allergy. The oral or rectal administration route is not recommended, especially for infants and children with gastroenteritis and young patients who often refuse to take the full dose of the drug because of its taste, for example. Therefore, the transdermal route of paracetamol administration is most useful and effective for pediatric applications [20,21]. Paracetamol is very slightly soluble in water but dissolves well in ethanol (50 mg/mL).

Paracetamol possesses poor skin permeability but it could be improved by adding ethanol or other enhancers, e.g., glyceryl oleate, PEG-40 stearate, or PEG-40 hydrogenated castor oil, which was observed for guar-based polymer [20]. Chitosan/guar gum hydrogels also proved effective in sustained paracetamol release [21]. Ethanol enhances the permeation of both polar and nonpolar molecules into the skin, but the mechanism of this process is not fully explained [22,23].

In this work, a new class of hybrid materials—polyurethane/Cloisite^®^ 30B (PU/PEO Cloisite^®^ 30B) nanocomposite hydrogel systems—for paracetamol drug delivery were studied. We present the swelling and release properties of these drug delivery systems depending on clay–Cloisite^®^ 30B concentration, as well as crosslinking agent type. Due to the complexity of the studied systems, a method of swelling and release processes modelling was extended, as well as a new, more versatile results interpretation being proposed, compared to our previous considerations in the article [18].

The main aim of this work was to comprehensively analyze the effect of nanofiller (Cloisite^®^ 30B) on the transport properties of the active substance in the proposed system. In the previous work, only one type of hydrogel was considered (PU/PEG 4000 TEA) for a 3% paracetamol concentration. In this work, we focus on the comparison of transport mechanism (swelling and release processes) for nanocomposite hydrogels, synthesized using different crosslinking agents (TEA, GLY), and we also answer the question of how different concentrations (1%, 5%) of paracetamol affect these processes. Thanks to this, it was possible to propose a new method of describing the transport process, taking into account the influence of crosslinking and the presence of nanoparticles on the efficiency of swelling and release of paracetamol from various nanocomposite matrices, especially the diffusion process description of short and long time approximation.

## 2. Materials, Synthesis, and Methods

### 2.1. Materials

Swelling and release studies were carried out for the following hydrogel nanocomposite matrices:(a)PU/PEG 4 000 TEA;(b)PU/PEG 4 000 TEA 0.25% Cloisite^®^ 30B;(c)PU/PEG 4 000 TEA 0.5% Cloisite^®^ 30B;(d)PU/PEG 4 000 TEA 1% Cloisite^®^ 30B;(e)PU/PEG 4 000 GLY;(f)PU/PEG 4 000 GLY 0.25% Cloisite^®^ 30B;(g)PU/PEG 4 000 GLY 0.5% Cloisite^®^ 30B.

In this notation, PU/PEG is the polymer nanocomposite, where PU is polyurethane and PEG is a polyethylene glycol with a molecular weight 4000 g/mol, Cloisite^®^ 30B (CLO) is organically modified with a quaternary ammonium salt sodium montmorillonite clay, purchased from Southern Clay Products Inc., Gonzales, TX, USA and presents in the matrices in 0%, 0.25%, 0.5%, 1% concentrations. GLY and TEA (Sigma-Aldrich Co.) represent crosslinking agents, which are glycerin and triethanolamine, respectively. The mechanical and structural properties of these matrices were described in our previous papers [7]. Swelling of paracetamol (PAR), from Sigma-Aldrich Co. (St. Louis, MO, USA), was carried out for 50% aqueous ethanol solutions (99.9% ethyl alcohol—P.P.H. “STANLAB”—diluted with demineralized water) in which 1% and 5% paracetamol solutions were prepared. Due to the limited solubility of paracetamol in aqueous ethanol solution, experiments were performed for the following mixtures:(a)1% of paracetamol in 50% water/ethanol solution;(b)5% of paracetamol in 50% water/ethanol solution.

### 2.2. Synthesis

Synthesis of nanocomposite PU/PEG hydrogels began with Cloisite^®^ 30B, which was dried for 6 h at 90 °C in a vacuum oven. Polyethylene glycol with a molecular weight of 4000 g/mol was dried in a vacuum oven (at lower pressure) at 90 °C for 30 min before use. 4,4′-methylene bis (cyclohexyl isocyanate) (HMDI), triethanolamine (TEA), glycerin (GLY), and acetone were used without further purification. First, a filler was dispersed for 15 min in a solution of PEG/acetone, using a homogenizer (8000 r.p.m.). Then, triethanolamine or glycerin was added to the solution as a modifier–crosslinking agent of low molecular weight. The selected crosslinking agents have a specific structure and do not contain additives that could affect the synthesis of the polyurethane systems. In addition, because of the relatively simple structure of crosslinking agents, we were able to adequately control the crosslinking process. Our laboratory experience with extenders containing hydroxyl groups (like butane-1,4-ol) and low molecular weight crosslinking compounds was the greatest. This is also particularly important in the context of appropriate substrate selection (proper determination of the ratio of NCO/OH groups) to obtain systems with the best possible properties, both thermal and mechanical. Finally, the appropriate amount of diisocyanate (HMDI) was dispersed in the mixture and was stirred at 80 °C for 1 min. Nanocomposites were poured into a round mold and dried at 40–50 °C for 24 h, until the acetone had evaporated [1]. The samples were marked depending on the amount of used nanofiller Cloisite^®^ 30B—0, 0.25, 0.5%, or 1%—and the type of crosslinker (GLY, TEA).

### 2.3. Methods

#### 2.3.1. Swelling Experiment—Gravimetric Method

Gravimetric measurements were performed using the RADWAG AS 110/C/2 laboratory scale (RADWAG, Radom, Poland) with an accuracy of ±0.001 g, at (20 ± 1) °C. Before tests, samples with a 10 mm diameter were cut and their thickness (from 1 to 1.2 mm ± 0.1 mm) was measured using a micrometer. The nanocomposite hydrogels were immersed in 5 mL of paracetamol/aqueous/ethanol solutions to absorb the solution. Matrices were weighed before swelling. The swelling process included an analysis of the weight changes over time. Initially, measurements were made every 2 min (for the first 20 min) and then the process was performed every 6 min. Before each weighing, the surface water was removed with a filter paper. The procedure was repeated until the equilibrium of swelling was reached. Swelling in % was calculated using Equation (1) with a measurement error of 5%.

#### 2.3.2. Mass Hysteresis Experiment—Drying Method

The mass hysteresis experiment was performed in three steps: (1) first swelling, (2) drying of the samples in the same intervals, (3) second swelling. Samples after the first swelling experiment were dried in a drying oven (Memmert GmbH + Co. KG, Schwabach, Germany, UNE 200 model) at 37 °C until materials were reswelled.

#### 2.3.3. Release Experiment—UV-Vis Method

Release experiments were carried out using a UV-Vis-type spectrophotometer (UV-2401 PC, Shimadzu, Tokyo, Japan). The matrices were immersed in 500 mL of deionized water, and then the paracetamol solution, released from the sample, was taken into a quartz cell and placed in a spectrophotometer. The absorption spectra of paracetamol released from nanocomposites were registered in the range of 200–350 nm. The amount of drug released from matrices was calculated based on the maximum absorbance of the recorded spectrum with a measurement error of 3%. These measurements were performed every 3 min until no changes in absorbance value were observed. After finishing the measurement and registration of the absorption spectrum, the solution from the cell was poured back into the beaker with the nanocomposite sample.

#### 2.3.4. Theoretical Background of the Swelling and Release Processes

The swelling kinetics of hydrogels are classified as a diffusion-controlled (Fickian) process, a relaxation-controlled Case II—time-independent transport—and anomalous (non-Fickian) diffusion. Fickian diffusion is observed when the diffusion rate is shorter than the polymer chain relaxation rate, but when it is far below the polymer chain relaxation rate, it is semi-Fickian diffusion, named “Less Fickian” behavior. When diffusion is much faster than the relaxation of polymer chains, Case II transport occurs. For anomalous diffusion, relaxation rates are comparable [24,25].

Swelling *S*(*t*) of the hydrogel is defined as the change in its mass over sorption time and is described by the following expression:(1)$$S\left(t\right)=\frac{M_{t}-M_{0}}{M_{0}}\cdot 100\;\left[\%\right],$$
where *M_0_* is the mass of dry hydrogel at initial time *t* = 0, and *M_t_* is the mass of the swollen hydrogel measured at time *t* [26,27].

A simple empirical power law model is used to determine time-dependent swelling of polymer:(2)$$S_{D}=k_{SD}t^{n},$$
where $S_{D}=M_{t}/M_{\infty }$ is a fractional uptake of solvent normalized to the equilibrium conditions, *k_SD_* is a swelling rate, and *n* is a diffusion exponent dependent on the sample geometry and transport mechanism of drug or solute [24,28]. Equation (2) is used to predict Fickian, Case II, anomalous, and Super Case II transport. The dependences of the *n* parameter on the transport mechanism for cylindrical systems are presented in Table 1 [29].

Transport in swelling systems is described by Fick’s Law—diffusion dependent on concentration gradient. For short time approximation, the solution to Fick’s equation is:(3)$$\frac{M_{t}}{M_{\infty }}=4\cdot {\left(\frac{Dt}{\pi l^{2}}\right)}^{\frac{1}{2}},$$
where *D* is the diffusion coefficient and *l* is the initial thickness of the sample. The square root of time dependence is characteristic of Fickian diffusion. Taking into account the viscoelastic behavior of polymer chains during swelling, Hopfenberg and Frisch [30] suggested that in polymers, sorption depends on relaxation (R) and diffusion (D) processes. Berens and Hopfenberg developed a diffusion–relaxation model [31] in which diffusion and relaxation components are separated:(4)$$M_{t}=M_{\infty ,SD}\left[1-\frac{6}{\pi^{2}}\sum_{n=1}^{x}\frac{1}{n^{2}}e^{-n^{2}k_{SD}t}\right]+\sum_{i}M_{\infty ,\;iSR}\left[1-e^{-k_{iSR}t}\right],$$
where $M_{\infty ,SD}$ is the equilibrium amount of absorbed solution in diffusion dependent process of swelling before relaxation, $k_{SD}$ is a diffusion rate constant, $M_{\infty ,SR}$ is an amount of penetrant uptake during relaxation process, $k_{SR}$ is a relaxation rate constant [28,31,32].

Release behavior of drugs from polymeric systems is described by a simple, semi-empirical equation:(5)$$\frac{M_{t}}{M_{\infty }}=k_{RD}t^{n},$$
where the *M*_t_ and *M*_∞_ are the amounts of the drug released at time *t* and infinite time *t*_∞_, *k_RD_* is a release rate, and *n* is the diffusion exponent characterizing the release mechanism [33]. This equation (Korsmeyer–Peppass model) describes the Fickian and non-Fickian release mechanism of swelling-controlled release of drugs or solutes from samples with different geometries: slabs, spheres, cylinders, and discs. In the case of Fickian release, the exponent *n* has the limiting values of 0.50, 0.45, and 0.43 for release from slabs, cylinders, and spheres, respectively. Equation (5) can be applied to the first 60% of the fractional release curves from thin slabs, assuming one-dimensional diffusion under perfect sink conditions [33]. Table 2 shows the ranges of parameter *n* related to the transport mechanism for various sample geometries.

The dominant molecular mechanism of release in swellable systems is diffusion and relaxation. In Case I (Fickian diffusion), solute release behavior in swelling-controlled release polymeric systems is described by diffusion coefficient *D*; in Case II transport by relaxation constant, non-Fickian (anomalous) behavior requires at least two parameters to describe both diffusion and relaxation processes. In the swelling-controlled hydrogel systems, an empirical power law modified by Peppas and Sahlin is used. This model takes into account the drug diffusion and polymer relaxation:(6)$$\frac{M_{t}}{M_{\infty }}=k_{DR}t^{n}+k_{RR}t^{2n},$$
where $k_{DR}=4\left[\frac{D}{\pi l^{2}}\right]$ and $k_{RR}=\frac{2k_{R}}{C_{0}l}\;$ are diffusion and relaxation (Case II) release rate constants, respectively; coefficient *n* is a Fickian diffusion exponent for samples of different geometrical shape that exhibit controlled release. If *k_RR_* is defined as the Case II relaxation constant, then the simple first-order kinetic equation can describe release process from slab:(7)$$\frac{M_{t}}{M_{\infty }}=1-e^{\left(-k_{RR}t\right)},$$
which is the consequence of the swelling-dependent relaxation process of polymer chains [25,33,34].

For one-dimensional, isothermal solute release from a thin polymer slab of initial thickness *l*, under perfect sink conditions and constant diffusion coefficient *D*, Fick’s second law solution for short-time approximation is given by the Equation (3) and is valid only for the first 60% of the total release processes $\left(0\leq \frac{M_{t}}{M_{\infty }}\leq 0.6\right)$. Late-time approximation is given by following equation:(8)$$\frac{M_{t}}{M_{\infty }}=1-\frac{8}{\pi^{2}}exp\left(-\frac{\pi^{2}Dt}{l^{2}}\right),$$
for $\left(0.4\leq \frac{M_{t}}{M_{\infty }}\leq 1\right)$ [33,35,36,37].

## 3. Results and Discussion

### 3.1. Swelling Studies

Swelling studies were performed for nanocomposite hydrogels PU/PEG 4 000 with different Cloisite^®^ 30B concentrations and crosslinking agents (TEA, GLY) in 1% and 5% of paracetamol in 50% water/ethanol solutions. Swelling curves for pure PU/PEG 4 000 hydrogels, synthesized using various crosslinking agents, are shown in Figure 1. For pure samples, without clay nanoparticles, for all paracetamol solutions, better swelling results were obtained for PU/PEG 4 000 TEA matrices than for PU/PEG 4 000 GLY systems. Compared to glycerin (GLY), the triethanolamine (TEA) molecule has a longer chain ended with OH groups, which creates a less packed hydrogel structure, which facilitates the sorption of solutions. Additionally, greater efficiency of this process is observed in ethanol solvents—about 400% more swelling than in water, about 300% for PU/PEG 4 000 TEA samples, and for PU/PEG 4 000 GLY about 120% in alcohol solutions, ~100% in water—because ethanol molecules have better chemical compatibility to PU/PEG 4 000 matrices. Differences in swelling properties of pure (without Cloisite^®^ 30B) hydrogel samples are shown in Figure 1.

Swelling curves for PU/PEG 4 000 TEA nanocomposites with Cloisite^®^ 30B nanoparticles are presented in Figure 2. The presence of Cloisite^®^ 30B nanoparticles in the PU/PEG 4 000 TEA matrix decreases (in comparison to pure hydrogels) the swelling of the system because nanoplatelets of clay form obstacles in the path of solvent diffusion in nanocomposite hydrogels (barrier effect). On the other hand, an increase in solvent swelling is observed in the system with a higher—0.5% concentration of Cloisite^®^ 30B, compared to nanocomposites containing 0.25% nanofiller. We suggest that this is a consequence of a process competing with the barrier effect—swelling of the clay. For samples containing 1% CLO, swelling reaches a lower value for water solvent—~200%—which is a consequence of the aggregation of clay nanoplatelets and decreasing free volumes in the system. On the other hand, the highest values (~450–550%) of swelling were registered for alcohol solutions because of additional swelling by aggregated Cloisite^®^ 30B nanoparticles. The presence of paracetamol does not significantly change the swelling process for 0.25 and 0.5% CLO content. However, for a 1% Cloisite^®^ 30B concentration, the swelling of paracetamol in a 50% aqueous ethanol solution decreases compared to pure water/ethanol solvent. This means that free volumes affected by aggregated clay limit PAR solution sorption to values obtained for 0.5% CLO concentration in the system.

Similar effects are observed for PU/PEG 4 000 GLY Cloisite^®^ 30B samples but the swelling of the system is much lower due to the greater degree of hydrogel crosslinking by GLY molecules and smaller free volumes in the system. The effectiveness of the swelling process is also influenced by the Cloisite^®^ 30B nanoparticle concentrations in the system and the type of solution. The presence of 0.25% concentration of Cloisite^®^ 30B nanoparticles in the PU/PEG 4 000 GLY matrix does not decrease swelling of the system because of the barrier effect, which was observed for PU/PEG 4 000 TEA samples. Swelling increases from 100% for pure hydrogel to 113% for PU/PEG 4 000 GLY 0.25% Cloisite^®^ 30B in water solvent and from about 120% to 165% for nanocomposites in alcohol solutions as a consequence of clay swelling. For samples containing 0.5% of CLO, swelling of the water and water/ethanol solutions decreases to ~100% and ~150%, respectively, as a result of the barrier effect and additional swelling of aggregated clay nanoplatelets. The presence of paracetamol does not significantly change the swelling processes. Swelling curves for PU/PEG 4 000 GLY nanocomposites with Cloisite^®^ 30B nanoparticles are present in Figure 3.

The theoretical analysis of the swelling process was performed by first fitting 60% of experimental curves using a simple empirical power law model, Equation (2), to obtain two diffusion parameters: *k*_SD_—diffusion swelling rate; *n*—diffusion exponent. The second part of the curves, describing the relaxation of polymer chains, was fitted using Equation (4). The parameters obtained from the theoretical fitting of swelling curves are presented in Table 3 and Table 4 for PU/PEG 4 000 TEA and PU/PEG 4 000 GLY nanocomposites, respectively. The fitting profiles are presented in Figure 4 and Figure 5.

Diffusion exponents, received from the fitting procedure to swelling curves for pure hydrogel matrices (PU/PEG 4 000 TEA), are *n* > 0.5, which means that anomalous diffusion occurs in these systems. For samples containing Cloisite^®^ 30B nanoparticles, the transport mechanism changes from anomalous to Fickian or “Less Fickian”. Exponent *n* obtained from fitting a mathematical model is not exactly equal to 0.5, as the theory indicates; its value is near or lower than 0.5. We experimentally observed that the exponent *n* may vary in the range of 0.50 to 0.55 values for PU/PEG 4 000 hydrogels with higher concentrations of nanoparticles only. We assume that this phenomenon, named “Near-Fickian” diffusion by us, occurs because in nanocomposites, there is a “double swelling” process related to the diffusion as a consequence of the concentration gradient and the swelling of nanoparticles, which affects the diffusion part of the transport mechanism of PAR in nanocomposites, but it does not yet affect the relaxation of polymer chains. The presence of nanoparticles, but also a higher concentration of paracetamol, reduces the time range of the diffusion swelling process. At 1% Cloisite^®^ 30B concentration in nanocomposite samples, “Less-Fickian” diffusion is observed, due to nanoparticle aggregation.

The presence of nanoparticles changes the nature of the swelling curve. In the case of systems without nanoparticles, the power law function was fitted to 40% of the fractional swelling curves, not to 60% as the theory assumes. The presence of a 1% concentration of nanoparticles reduces this value to 23%. Nanoparticles, as well as polymer degrees of crosslinking, significantly limit free diffusion in nanocomposites.

Diffusion swelling rates for paracetamol solutions, as well as maximum swelling values, depend on Cloisite^®^ 30B concentration in the systems and are higher for samples containing nanoparticles. The highest value of *S*_max_ was obtained for PU/PEG 4 000 TEA 1% Cloisite^®^ 30B (454%), and the lowest value of *S*_max_ was observed for pure PU/PEG 4 000 TEA (350%). The relaxation rate decreased for 0.5% CLO concentration in the sample and increased for 1% CLO in comparison to pure systems due to barrier effect and swelling of the clay, respectively. The relaxation rate increases slightly with increasing paracetamol concentration in the presence of nanoparticles and decreases when they are not present in the system because clays promote the expansion of the hydrogel matrix and accelerate the diffusion process. The relaxation swelling process is prolonged and the diffusion swelling process is shortened during all sorption time ranges.

As can be seen in Table 4, the swelling exponents *n* are greater than 0.5 for the PU/PEG 4 000 GLY hydrogels, characterized by a higher degree of crosslinking than those containing TEA. It means that non-Fickian, anomalous diffusion occurs for all samples. The presence of 0.5% of Cloisite^®^ 30B nanoparticles in the system additionally inhibits the sorption process due to the barrier effect. The swelling rates (*k*_SR_ and *k*_SD_) as well as maximum swelling value (*S*_max_) are lower than those observed for PU/PEG 4 000 GLY with a 0.25% clay concentration.

The swelling exponents (*n*~0.5) obtained from fitting the swelling curves with a power law model using Equation (2) for PU/PEG 4 000 TEA samples containing Cloisite^®^ 30B nanoparticles indicate a diffusion-dependent swelling process. Therefore, it is possible to determine the diffusion coefficients for these processes using Equation (3). The fits of the swelling curves and the obtained diffusion coefficients are presented in Figure 6 and Table 5. Diffusion coefficients are higher for matrices containing 0.5% of Cloisite^®^ 30B (1.58·10^−7^ cm^2^/s—1% of paracetamol solution; 1.89·10^−7^ cm^2^/s—5% of paracetamol solution) than for 1% clay content (0.96·10^−7^ cm^2^/s—1%, 5% of paracetamol solution) because in these systems, there is a barrier effect enhanced by the additional aggregation of Cloisite^®^ 30B nanoparticles.

### 3.2. Swelling Hysteresis Studies

Swelling hysteresis curves were measured for PU/PEG 4 000 TEA pure sample and PU/PEG 4 000 TEA 0.5% Cloisite^®^ 30B, PU/PEG 4 000 1% Cloisite^®^ 30B nanocomposites in 1% and 5% of paracetamol in 50% ethanol/water solutions. The mass of hydrogels was measured at appropriate time intervals during the three processes: (1) swelling of the paracetamol solution by matrices, (2) drying hydrogels at constant temperature (37 °C), and (3) swelling of demineralized water by dried samples. The obtained hysteresis curves for PU/PEG 4 000 TEA-type hydrogels are presented in Figure 7.

The swelling and drying curves obtained for all measured samples have different courses. The presence of nanoparticles in the matrices causes significant changes in the course of these curves. For pure hydrogels swelling curves are similar in the diffusion part of this process and different in the relaxational part of the swelling process. The Cloisite^®^ 30B nanoparticle’s presence in the system causes changes in the time range of the diffusion swelling process. The same course of swelling curves is observed in the different time ranges depending on nanoparticle concentration. For pure samples, time ranges do not change up to ~60 and ~100 min of sorption time, ~55 and ~60 min for 0.5% Cloisite^®^ 30B systems, and ~10 and ~0 min for 1% Cloisite^®^ 30B content in nanocomposites for 1% and 5% PAR concentration, respectively. The same values of *S* parameters obtained for both the first and second swelling experiments decrease with increasing Cloisite^®^ 30B nanoparticles concentration; about 50% of the swelling curve for 0.5% Cloisite^®^ 30B and 10% for 1% concentration of clay nanoplatelets. In the second swelling course, the *S*_max_ parameters reach a lower value than that obtained for the first one, which is because during the drying process, only water or ethanol solutions were removed from the system, and paracetamol remained confined in the polymer matrix. Hence, the second swelling process is less effective due to polymer chains reorganization under the influence of temperature.

For higher concentrations of nanoparticles in the system, more pronounced differences in the maximum value of swelling are observed—71% and 118% for PU/PEG 4 000 TEA 1% Cloisite^®^ 30B in 1% and 5% of paracetamol in water/ethanol solution, respectively. For 0.5% concentration of nanoparticles, these differences are smaller than for 53% and 43% for 1% and 5% of paracetamol water/ethanol solutions compared with 86% and 49% pure samples, respectively.

The same analysis was performed for the PU/PEG 4 000 GLY matrices type and similar results were obtained (Figure 8). First and second swelling curves are different courses, and the maximum value of swelling differences are higher for samples with Cloisite^®^ 30B nanoparticles. For pure hydrogels, swelling curves are similar in the diffusional part of this process and different in the relaxational part of the swelling process. The Cloisite^®^ 30B nanoparticles’ presence in the system causes changes in the time range of the diffusion swelling process. The same course of swelling curves are observed in the different time ranges depending on nanoparticles and paracetamol presence in the system. For pure samples, time ranges do not change up to ~10 and ~40 min of sorption time and are the same up to ~8 min for 0.25 and 0.5% Cloisite^®^ 30B concentrations in the systems for 1% and 5% PAR concentration, respectively. In the second swelling, the *S*_max_ parameters reach values less than for the first one. The largest differences are observed for 0.25% concentration of Cloisite^®^ 30B in the system—93% in PU/PEG 4 000 GLY 0.25% Cloisite^®^ 30B in 5% of paracetamol in water/ethanol solution. The PU/PEG 4 000 GLY 0.5% Cloisite^®^ 30B samples show smaller differences in *S*_max_ values (about 50%) than pure matrices (81% and 86% for 1% and 5% of paracetamol aqueous ethanol solutions).

Analyzing the PU/PEG 4 000 GLY matrices, it can be seen that the drying process led to an almost complete release of the water, water/ethanol solution. In the second swelling process, smaller values of swelling were achieved (up to approximately 80%) than in the case of the first swelling. The samples containing 0.25% Cloisite^®^ 30B released almost all of the solution during the drying process, about 1% of the solution remained inside the samples, and for PU/PEG 4 000 GLY 0.5% Cloisite^®^ 30B matrix, about 7%. Based on the hysteresis analysis, it can be concluded that the polymer chains were not destroyed during the first swelling and drying processes, and it is possible that there was a polymer chain reorganization caused by Cloisite^®^ 30B clay nanoparticle presence, accelerated by higher temperature.

Drying curves are similar, nonlinear, one-stage courses for all studied samples; it is not possible to distinguish diffusional and relaxational processes of solution release. Based on this, it can be concluded that the paracetamol that remains in the matrix was partially bound to it by hydrogen bonds during the first swelling process [9].

The observed changes in the swelling hysteresis are smaller for PU/PEG 4 000 GLY-type hydrogels than for PU/PEG 4 000 TEA-type due to higher crosslinking of the system with the GLY crosslinking agent.

### 3.3. Release Process of the Paracetamol—Spectroscopic Studies

The release process of the paracetamol from nanocomposite matrices was also studied using a steady-state UV-Vis spectroscopic technique. Figure 9 shows example absorption spectra of paracetamol released from PU/PEG 4 000 TEA- and GLY-type hydrogels. The analysis was performed for the band with an absorption maximum of 246 nm, which is characteristic of the paracetamol molecule. The released mass of the PAR was calculated from the maximum absorbance value within a given release time *t,* using data obtained for standard solutions.

Two main physical phenomena responsible for the release of the paracetamol from hydrogels are diffusion and swelling. These processes depend on the system geometry and the structure of the polymer matrix. The diffusional release process is driven by the presence of a gradient of PAR concentration and is effective over short distances. Relaxational release is observed as a consequence of relaxation of tangled polymeric chains during swelling. Therefore, these two processes were separated and fitted using appropriate models in short ($0\leq \frac{M_{t}}{M_{\infty }}\leq 0.6$) and long ($0.4\leq \frac{M_{t}}{M_{\infty }}\leq 1$) time ranges on PAR release curves.

Release profiles of paracetamol for PU/PEG 4 000 TEA (Figure 10) and PU/PEG 4 000 GLY (Figure 11) hydrogels were plotted from the molar absorption coefficient value at λ = 246 nm in release time for all studied matrices. Based on these profiles, Equation (5) was applied to the first 60% of the fractional release curves to estimate diffusion release rate (*k*_RD_) and release exponent (*n*). Using the spectral data, the relaxational release rate (*k*_RR_) was determined using Equation (7). The values of parameters describing the release of paracetamol from matrices, obtained based on fitting mathematical models to experimental data, are presented in Table 6 and Table 7. The analysis of these parameters allowed us to determine the transport mechanism of paracetamol from the differently crosslinked hydrogel matrices in the presence or absence of clay nanoparticles.

The analysis of diffusion and relaxation release rate coefficients show that these processes are most effective for PU/PEG 4 000 TEA samples with Cloisite^®^ 30B nanoparticles. Nanoplatelets cause additional loosening of the polymer chain structure, facilitating the diffusion process. The determined diffusion exponent values (*n* ≈ 0.5) indicate the near-Fickian diffusion or less-Fickian transport mechanisms for samples containing Cloisite^®^ 30B clay. Anomalous diffusion (non-Fickian diffusion) is observed in pure hydrogels (*n* > 0.5).

Additionally, a decrease in the amount of the released active substance is observed with an increase in clay concentration in the system. This is related to the barrier effect and the interaction of PAR molecules with matrix and nanoparticles, as well as depending on the concentration of paracetamol in hydrogels. When the polymer swells, the diffusion pathlength increases cause a decrease in the drug concentration gradient, and thus decreasing drug release rates [38]. For a higher concentration of paracetamol, its significant release is observed, but for 1% of Cloisite^®^ 30B in the nanocomposite, the maximum mass of released paracetamol decreases due to the competitive barrier effect. The release relaxation rate increases with increasing CLO concentration, which means that the nanoparticles increase the efficiency of untangling polymer chains during the swelling process, but also because of the more pronounced swelling properties of clay nanoparticles. When polymer mobility increases, drug mobility increases, and the release rate also increases.

Release exponents obtained from release profiles of PU/PEG 4 000 GLY-type hydrogels indicate anomalous diffusion for pure samples and less-Fickian for those containing Cloisite^®^ 30B clay nanoplatelets. Diffusion and relaxation release rates are higher for samples containing CLO nanoparticles in comparison to pure PU/PEG 4 000 GLY matrix. In the case of 0.5% Cloisite^®^ 30B concentration in the system, the diffusion rate decreases due to the barrier effect but the relaxation rate increases—nanoplatelets affect polymer chains. Due to the barrier effect, the mass of released paracetamol decreases with the increase in CLO concentration in nanocomposite. The diffusion rate is more effective (faster) than the relaxation rate. This phenomenon is presented in Figure 12.

The analysis of release exponents allowed us to identify materials in which diffusion is the dominant transport mechanism. For these materials, the diffusion coefficients were determined using Equation (3) for short time approximation and Equation (8) for long time approximation. Calculations were also performed for systems for which the *n* exponent was close to the 0.5 value. The results of fitting the mathematical models to the experimental data are presented in Figure 13 and Table 8 for PU/PEG 4 000 TEA- and GLY-type nanocomposites.

The diffusion coefficients of paracetamol molecule, calculated for PU/PEG 4 000 TEA nanocomposites, increase with increasing concentration (0.5, 1%) of Cloisite^®^ 30B in matrices for 0.5% of PAR water/ethanol solution and decrease for 1% PAR solution both for short time and long time approximation parts of the curve.

Diffusion coefficients *D*_short_app_ and *D*_long_app_ are of the same order and are higher for long time approximation.

The PAR diffusion coefficients, determined for PU/PEG 4 000 GLY nanocomposites, increase with increasing concentration (0, 0.25, 0.5%) of Cloisite^®^ 30B in matrices. Its values are of the same order for short time and long time approximation parts of the curve. For 0 and 0.25% CLO concentrations, *D*_short_app_ reaches a lower value than the *D*_long_app_ coefficient; however, for 0.5% Cloisite^®^ 4 000 concentration, this value is higher for the long time approximation part of the release profile.

## 4. Conclusions

The transport mechanism of the paracetamol molecule was determined in PU/PEG 4 000 TEA- and PU/PEG 4 000 GLY-type hydrogels containing Cloisite^®^ 30B clay nanoparticles. Swelling and release processes of active substance were studied using gravimetric and spectroscopic (UV-Vis) techniques, respectively. The analysis of swelling and release curves allowed us to determine the transport mechanism in polymer matrices based on the parameters (swelling and release rates, diffusion coefficients, swelling and release exponent), obtained by fitting appropriate mathematical models. The influence of the concentration of Cloisite^®^ 30B nanoplatelets and paracetamol on the release and swelling parameters was examined for matrices with various degree of crosslinking.

Analyzing the data obtained from the steady-state spectroscopic and gravimetric measurements, one can state that:The presence of Cloisite^®^ 30B nanoparticles in the matrix decreases swelling of the system because of the barrier effect in comparison to the pure one (without clay nanoparticles);A higher concentration of nanoparticles causes an increase in swelling as a result of “double swelling” behavior—competitive process with the barrier effect;Exceeding a certain critical nanofiller concentration value (0.5%) leads to its aggregation, which causes decreasing free volumes in the system and decreasing swelling in the case of more crosslinked PU/PEG 4 000 GLY systems;Relaxation rates decrease due to the barrier effect and increase due to the swelling of the aggregated clay; the relaxation rate increases slightly with increasing paracetamol concentration in the presence of nanoparticles and decreases when they are not present in the system because clays promote swelling of the polymer matrix;Release relaxation rates increase with increasing CLO concentration; the nanoparticle presence in the system increases the efficiency of hydrogel expansion during the swelling processes, also due to clay–matrix interaction;In the case of too high a concentration of Cloisite^®^ 30B, the diffusion rate decreases due to the barrier effect correlated with clay aggregation, because the diffusion pathlength increases and causes a decrease in the drug concentration gradient;Diffusion coefficients of paracetamol molecules in the release process increase with increasing concentration of Cloisite^®^ 30B in all matrices; diffusion coefficients *D*_short_app_ and *D*_long_app_ are of the same order and are higher for long time approximation;In matrices characterized by a high degree of crosslinking, anomalous diffusion occurs; the presence of Cloisite^®^ 30B nanoparticles in the system additionally inhibits the sorption process due to the barrier effect; the swelling rates, as well as the maximum swelling value, are lower than those observed for lower degrees of crosslinking systems.

The theoretical analysis of the swelling and release processes showed that for pure hydrogels, an anomalous diffusion transport mechanism occurs and changes to “Near-Fickian” or “Less Fickian” diffusion for matrices with Cloisite^®^ 30B filler. In this work, it would be proposed to name the “Near Fickian” diffusion process for which the swelling and release exponent *n* is in the range of 0.50–0.55 due to a “double swelling” phenomenon related to the diffusion as a consequence of the concentration gradient and swelling of nanoparticles.

The analysis of transport mechanisms in polyurethane nanocomposites containing Cloisite^®^ 30B nanoparticles showed that the drug release process can be controlled by the concentration of nanoparticles in the system and the degree of crosslinking of the polymer matrix. The presence of paracetamol does not significantly change the swelling process; however, the matrix degree of crosslinking increases, resulting in a decrease in its swelling. This can be used to produce new patches with a controlled drug release process.

## Figures and Tables

**Figure 1 materials-17-00040-f001:**
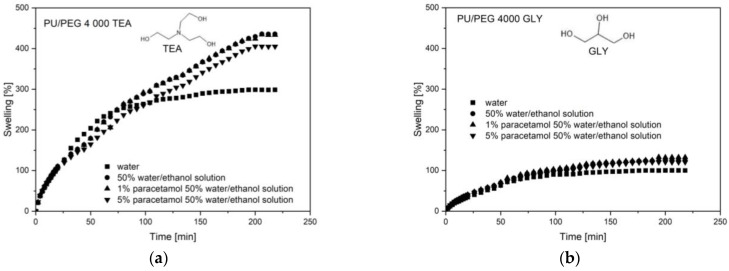
Swelling curves of (**a**) PU/PEG 4 000 TEA and (**b**) PU/PEG 4 000 GLY pure hydrogel matrices.

**Figure 2 materials-17-00040-f002:**
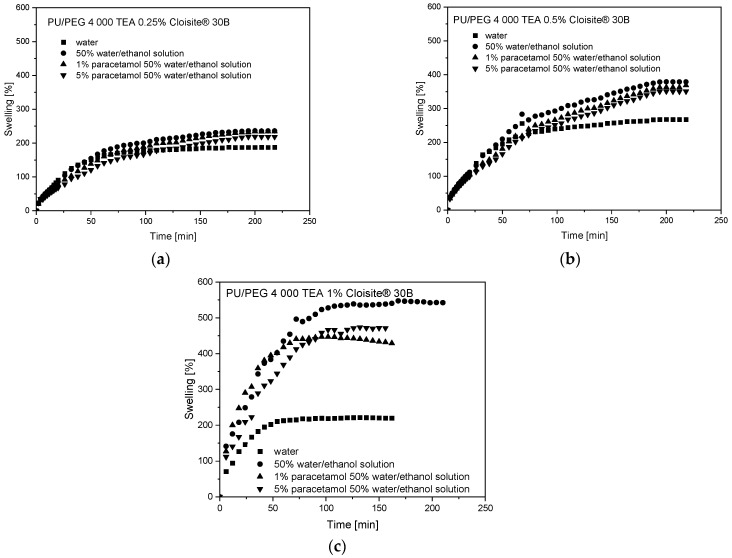
Swelling curves of (**a**) PU/PEG 4 000 TEA 0.25% Cloisite^®^ 30B, (**b**) PU/PEG 4 000 TEA 0.5% Cloisite^®^ 30B, and (**c**) PU/PEG 4 000 TEA 1% Cloisite^®^ 30B hydrogel matrices.

**Figure 3 materials-17-00040-f003:**
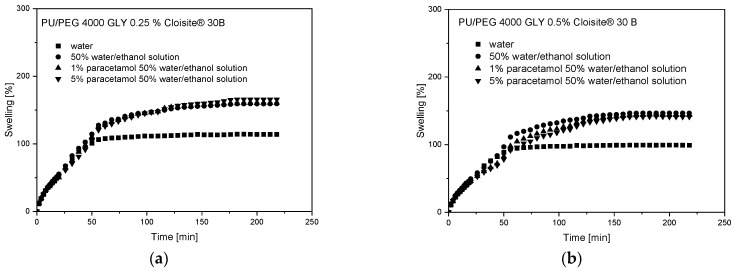
Swelling curves of (**a**) PU/PEG 4 000 GLY 0.25% Cloisite^®^ 30B and (**b**) PU/PEG 4 000 GLY 0.5% Cloisite^®^ 30B hydrogel matrices.

**Figure 4 materials-17-00040-f004:**
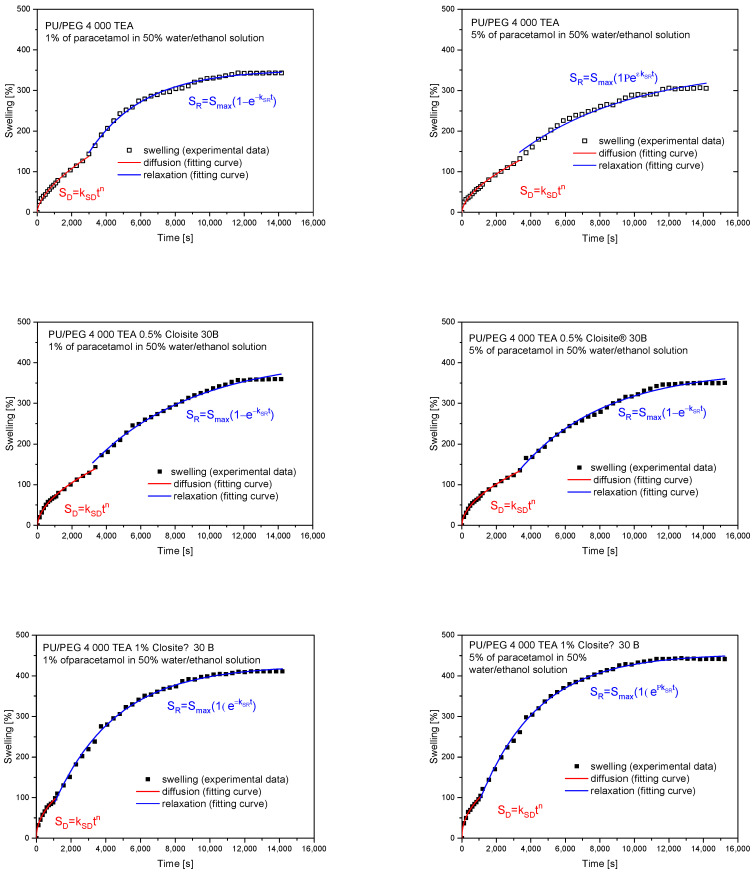
Swelling curve fitting for PU/PEG 4 000 TEA nanocomposites.

**Figure 5 materials-17-00040-f005:**
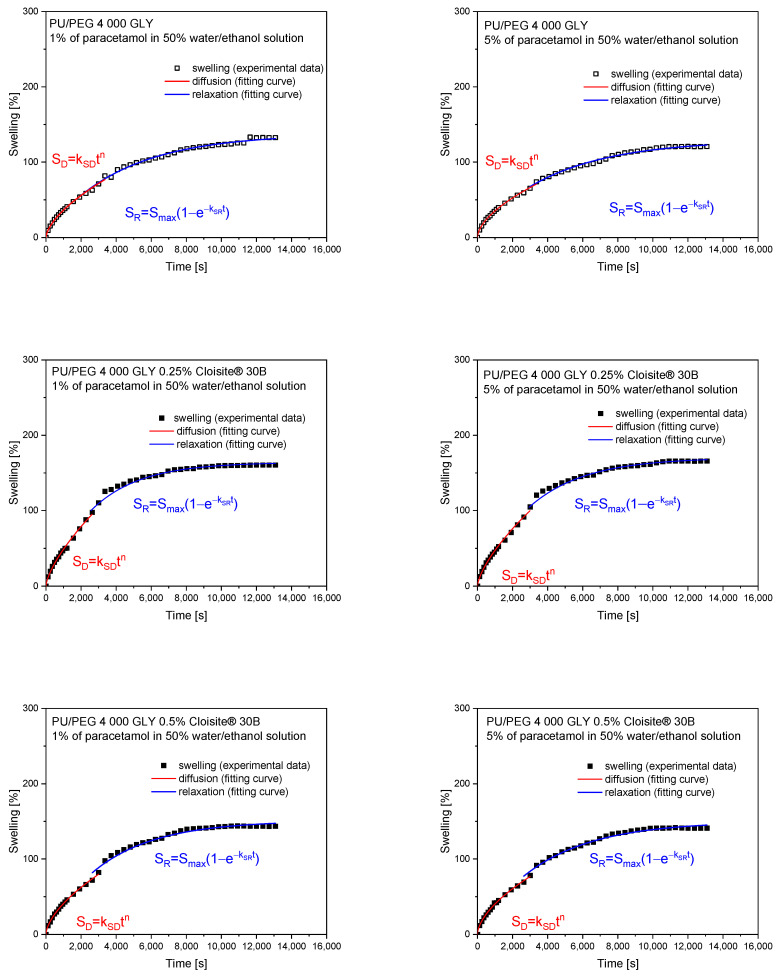
Swelling curve fitting for PU/PEG 4 000 GLY nanocomposites.

**Figure 6 materials-17-00040-f006:**
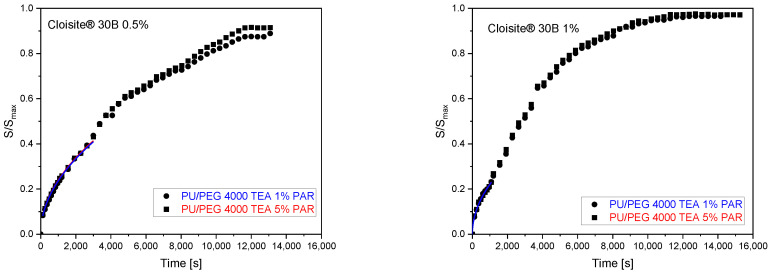
Determination of diffusion coefficients from swelling curves for PU/PEG 4 000 TEA Cloisite^®^ 30B nanocomposites.

**Figure 7 materials-17-00040-f007:**
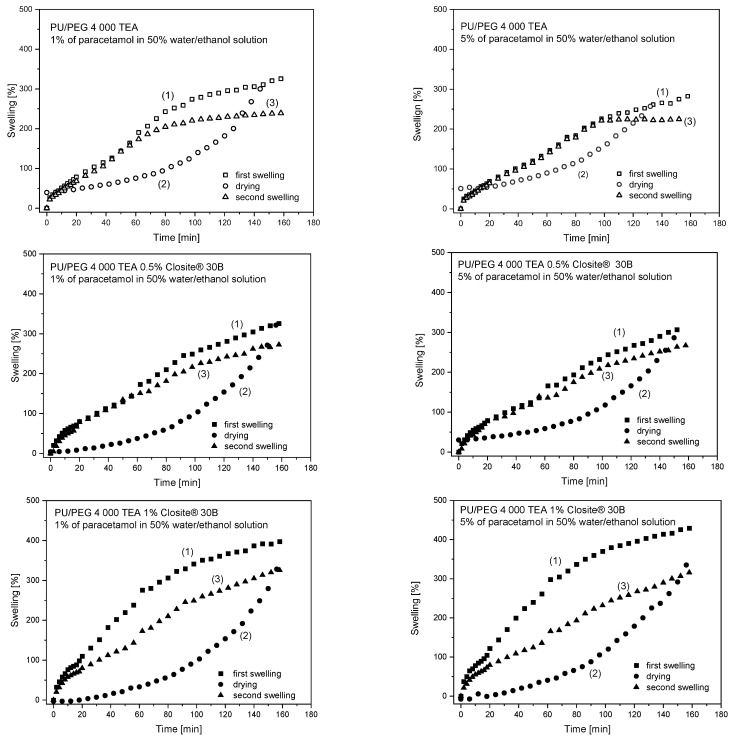
Swelling hysteresis curves for PU/PEG 4 000 TEA type hydrogels.

**Figure 8 materials-17-00040-f008:**
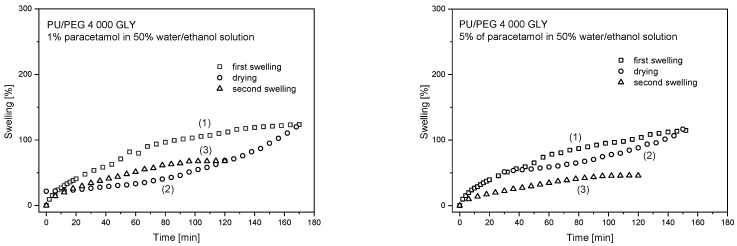
Swelling hysteresis curves for PU/PEG 4 000 GLY-type hydrogels.

**Figure 9 materials-17-00040-f009:**
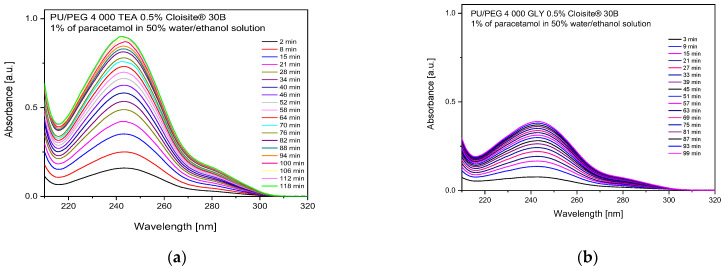
Absorption spectra of paracetamol registered during the release process from (**a**) PU/PEG 4 000 TEA- and (**b**) GLY-type matrices.

**Figure 10 materials-17-00040-f010:**
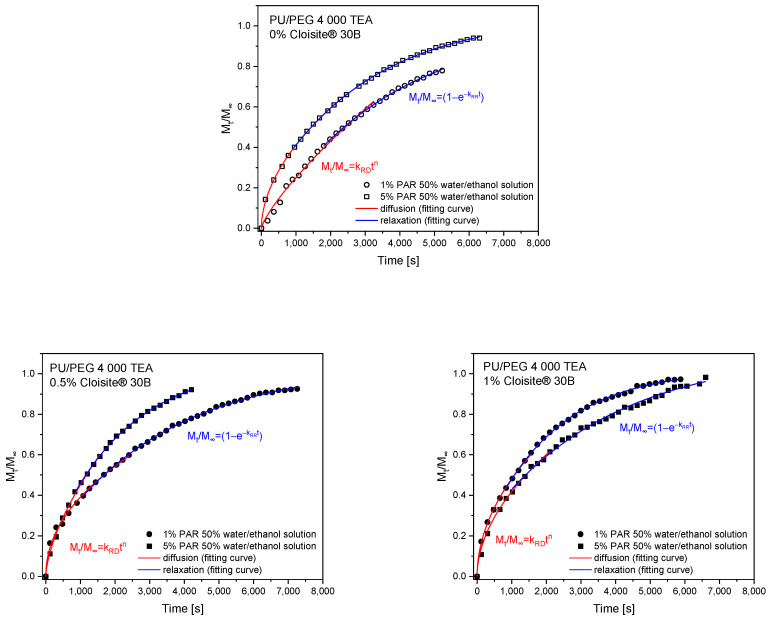
Release profiles of paracetamol from PU/PEG 4 000 TEA-type matrices fitted with mathematical models.

**Figure 11 materials-17-00040-f011:**
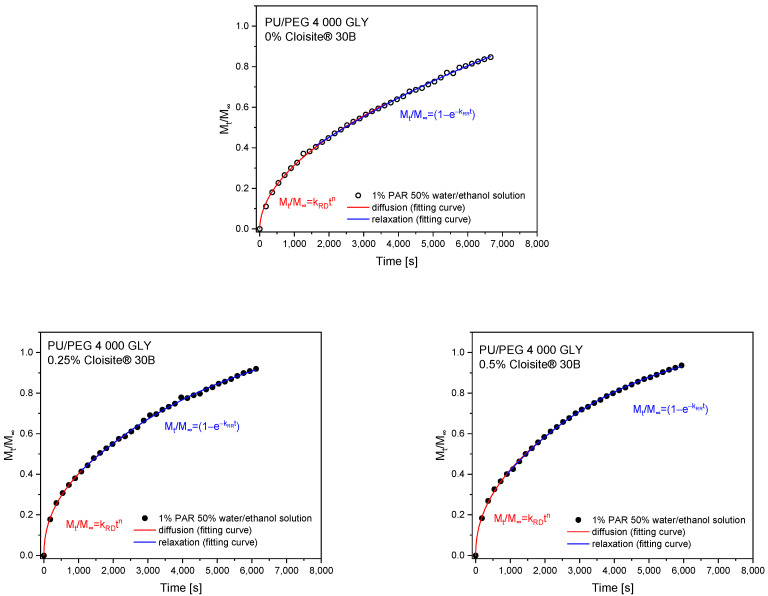
Release profiles of paracetamol from PU/PEG 4 000 GLY-type matrices fitted with mathematical models.

**Figure 12 materials-17-00040-f012:**
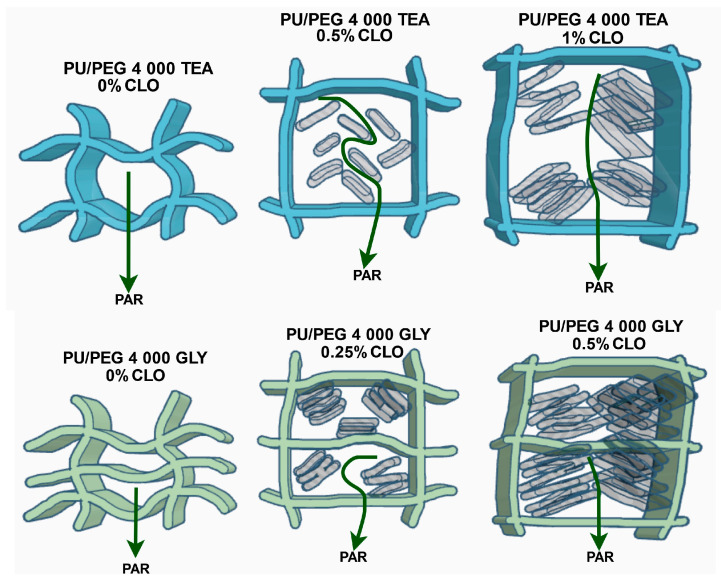
Schematical model of “double swelling”, barrier effect, and aggregation of clay—influence on paracetamol release process.

**Figure 13 materials-17-00040-f013:**
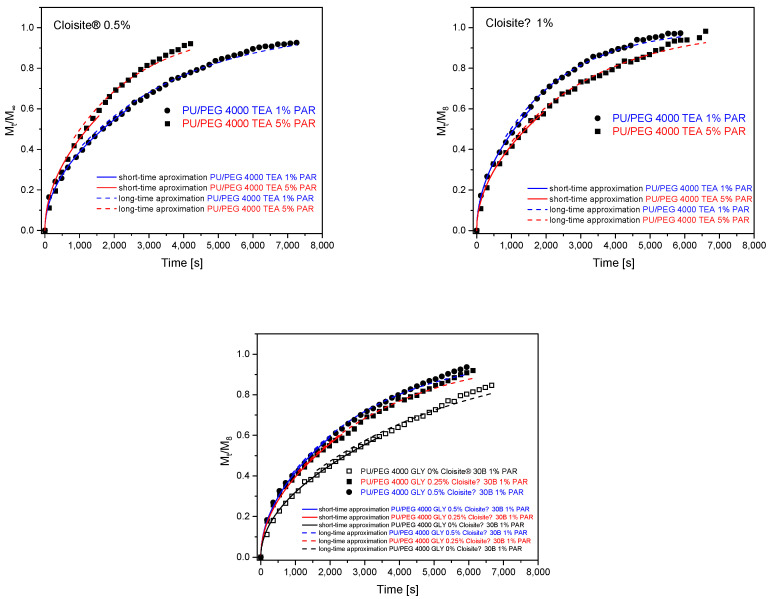
Determination of diffusion coefficients from release curves for PU/PEG 4 000 TEA- and PU/PEG 4 000 GLY-type hydrogels.

**Table 1 materials-17-00040-t001:** Diffusion exponent *n* and swelling dependent transport mechanism in thin polymeric systems (slabs) [29].

Diffusion Exponent *n*	Transport Mechanism
*n* < 0.5	Less Fickian diffusion
*n* = 0.5	Fickian diffusion
0.5 < *n* < 1	Anomalous diffusion
*n* = 1	Case II transport
*n* > 1	Super Case II transport

**Table 2 materials-17-00040-t002:** Diffusion exponent *n* and transport mechanism for swellable controlled release systems [33].

Diffusion Exponent *n*			Transport Mechanism
Slab	Cylinder	Sphere	
0.5	0.45	0.43	Fickian diffusion (Case I)
0.5 < *n* < 1.0	0.45 < *n* < 0.89	0.43 < *n* < 0.85	Non-Fickian diffusion
1.0	0.89	0.85	Case II transport (zero-order release)

**Table 3 materials-17-00040-t003:** Swelling parameters for PU/PEG 4 000 TEA nanocomposites.

	PU/PEG 4 000 TEA					
	50% Water/Ethanol Solution					
**CLO (%)**	0	0	0.5	0.5	1	1
**PAR (%)**	1	5	1	5	1	5
*** *k*_SD_^′^ (s^−n^)**	1.04 ± 0.13	1.08 ± 0.12	1.64 ± 0.13	1.73 ± 0.08	3.36 ± 0.40	4.13 ± 0.38
** *n* **	0.61 ± 0.02	0.59 ± 0.01	0.55 ± 0.01	0.54 ± 0.01	0.48 ± 0.02	0.46 ± 0.01
***S*_max_ (%)**	350.96 ± 1.97	360.48 ± 7.45	414.81 ± 7.76	383.17 ± 5.13	426.09 ± 2.17	454.39 ± 1.77
** *k* _SR_ ** **(·10^−4^ s^−1^)**	3.16 ± 0.11	1.49 ± 0.07	1.65 ± 0.09	2.00 ± 0.11	2.78 ± 0.07	2.92 ± 0.06
**Transport mechanism**	Non-Fickian diffusion	Non-Fickian diffusion	Near-Fickian diffusion	Near-Fickian diffusion	Less-Fickian diffusion	Less-Fickian diffusion

* *k*_SD_^’^ parameter included *S*_max_.

**Table 4 materials-17-00040-t004:** Swelling parameters for PU/PEG 4 000 GLY nanocomposites.

	PU/PEG 4 000 GLY					
	50% Water/Ethanol Solution					
**CLO (%)**	0	0	0.25	0.25	0.5	0.5
**PAR (%)**	1	5	1	5	1	5
***k*_SD_^′^ (s^−n^) ***	0.50 ± 0.06	0.66 ± 0.05	0.42 ± 0.05	0.44 ± 0.05	0.62 ± 0.04	0.71 ± 0.05
** *n* **	0.62 ± 0.02	0.58 ± 0.01	0.69 ± 0.02	0.68 ± 0.02	0.61 ± 0.01	0.59 ± 0.01
***S*_max_ (%)**	137.67 ± 1.38	129.49 ± 1.40	164.06 ± 1.19	170.26 ± 1.32	151.04 ± 1.44	150.12 ± 1.39
***k*_SR_ (** **·10^−4^ s^−1^)**	2.30 ± 0.07	2.21 ± 0.07	3.62 ± 0.01	3.15 ± 0.10	2.84 ± 0.10	2.57 ± 0.08
**Transport mechanism**	Non-Fickian diffusion	Non-Fickian diffusion	Non-Fickian diffusion	Non-Fickian diffusion	Non-Fickian diffusion	Non-Fickian diffusion

* *k*_SD_^’^ parameter included *S*_max_.

**Table 5 materials-17-00040-t005:** Diffusion coefficients for PU/PEG 4 000 TEA Cloisite^®^ 30B nanocomposites.

Sample	Solution	*l* (cm)	*D* (cm^2^·s^−1^·10^−7^)
**PU/PEG 4 000 TEA** **0.5% Cloisite^®^ 30B**	1% PAR in 50% water/ethanol solution	0.12	1.58 ± 0.03
	5% PAR in 50% water/ethanol solution	0.13	1.89 ± 0.03
**PU/PEG 4 000 TEA** **1% Cloisite^®^ 30B**	1% PAR in 50% water/ethanol solution	0.10	0.96 ± 0.02
	5% PAR in 50% water/ethanol solution	0.10	0.96 ± 0.02

**Table 6 materials-17-00040-t006:** Release parameters for PU/PEG 4 000 TEA-type hydrogels.

PU/PEG 4 000 TEA 50% Water/Ethanol Solution	0% Cloisite^®^ 30B		0.5% Cloisite^®^ 30B		1% Cloisite^®^ 30B	
	1% PAR	5% PAR	1% PAR	5% PAR	1% PAR	5% PAR
***k*_RD_ (s^−n^)**	0.0012 ± 0.0002	0.0105 ± 0.0005	0.0148 ± 0.0015	0.0056 ± 0.0006	0.0147 ± 0.0009	0.0112 ± 0.0019
** *n* **	0.78 ± 0.02	0.53 ± 0.01	0.47 ± 0.01	0.64 ± 0.01	0.51 ± 0.01	0.52 ± 0.02
***k*_RR_ (·10^−4^ s^−1^)**	2.81 ± 0.15	3.36 ± 0.04	2.89 ± 0.06	4.62 ± 0.13	4.82 ± 0.11	2.77 ± 0.18
***M*_∞_ (g)**	0.0212 ± 0.0003	0.1825 ± 0.0025	0.0563 ± 0.0009	0.0818 ± 0.0011	0.0476 ± 0.0005	0.1015 ± 0.0020
**Transport** **mechanism**	Non-Fickian diffusion	Non-Fickian diffusion	Less-Fickian diffusion	Non-Fickian diffusion	Near-Fickian diffusion	Near-Fickian diffusion

**Table 7 materials-17-00040-t007:** Release parameters for PU/PEG 4 000 GLY-type hydrogels.

PU/PEG 4 000 GLY 50% Water/Ethanol	0% Cloisite^®^ 30B	0.25% Cloisite^®^ 30B	0.5% Cloisite^®^ 30B
	1% PAR	1% PAR	1% PAR
***k*_RD_ (s^−n^)**	0.0085 ± 0.0004	0.0173 ± 0.0006	0.0166 ± 0.0009
** *n* **	0.52 ± 0.02	0.46 ± 0.01	0.47 ± 0.01
***k*_RR_ (·10^−4^ s^−1^)**	1.01 ± 0.09	2.13 ± 0.12	2.74 ± 0.06
***M*_∞_ (g)**	0.0303 ± 0.0011	0.0253 ± 0.0007	0.0236 ± 0.0006
**Transport** **mechanism**	Non-Fickian diffusion	Less-Fickian diffusion	Less-Fickian diffusion

**Table 8 materials-17-00040-t008:** Diffusion coefficients for PU/PEG 4 000 TEA- and PU/PEG 4 000 GLY-type hydrogels—release process.

Sample	Solution	*l* (cm)	*D*_short_app_ (cm^2^·s^−1^·10^−7^)	*D*_long_app_ (cm^2^·s^−1^·10^−7^)
**PU/PEG 4 000 TEA** **0.5% Cloisite^®^ 30B**	1% PAR 50% water/ethanol solution	0.12 ± 0.01	4.43 ± 0.07	4.52 ± 0.04
	5% PAR 50% water/ethanol solution	0.12 ± 0.01	5.80 ± 0.26	6.94 ± 0.13
**PU/PEG 4 000 TEA** **1% Cloisite^®^ 30B**	1% PAR 50% water/ethanol solution	0.12 ± 0.01	6.54 ± 0.05	7.18 ± 0.08
	5% PAR 50% water/ethanol solution	0.12 ± 0.01	4.99 ± 0.13	5.30 ± 0.07
**PU/PEG 4 000 GLY** **0% Cloisite^®^ 30B**	1% PAR 50% water/ethanol solution	0.11 ± 0.01	2.43 ± 0.02	2.63 ± 0.04
**PU/PEG 4 000 GLY** **0.25% Cloisite^®^ 30B**		0.11 ± 0.01	3.72 ± 0.05	3.85 ± 0.04
**PU/PEG 4 000 GLY** **0.5% Cloisite^®^ 30B**		0.11 ± 0.01	4.16 ± 0.05	4.32 ± 0.05

## Data Availability

Data are contained within the article.

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
