# Peer review of "Transport Mechanism of Paracetamol (Acetaminophen) in Polyurethane Nanocomposite Hydrogel Patches—Cloisite^®^ 30B Influence on the Drug Release and Swelling Processes"

_materials, 2023, doi:10.3390/ma17010040_

Round 1
Reviewer 1 Report
Comments and Suggestions for Authors
Dear Editor,
I have had the opportunity to review the manuscript entitled "Transport Mechanism of Paracetamol in Polyurethane Nanocomposite Hydrogel Patches – Cloisite® 30B Influence on Drug Release and Swelling Processes" submitted to Materials. The study investigates a relevant area concerning the release of paracetamol and the influence of Cloisite® 30B on drug release and swelling processes in polyurethane nanocomposite hydrogel patches. While the manuscript presents valuable insights, there are certain aspects that require further consideration before acceptance.
-
Mechanical Properties and Physical Intercalation: One notable limitation of the manuscript is the absence of information on the mechanical properties and physical intercalation of Cloisite® 30B in the polyurethane nanocomposite hydrogel patches. The authors should clarify whether Cloisite® 30B is intercalated or dispersed as macroscopic particles and investigate the influence of this phyllosilicate on the mechanical properties. A thorough viscoelastic analysis, coupled with TEM microscopy, would be beneficial to understand the material's behavior more comprehensively.
-
Formal Adjustments: The manuscript should undergo formal adjustments regarding the expression of molecular weights of polymers. It is recommended to present the molecular weight in Daltons (Da) or without dimensions to ensure clarity and conformity to standard conventions.
In conclusion, while the manuscript addresses an important aspect of drug delivery and nanocomposite hydrogel patches, the aforementioned considerations should be addressed to enhance the completeness and quality of the research. I recommend inviting the authors to revise the manuscript to incorporate these aspects before reconsidering it for publication.
Literature update: The literature panorama should be taken into consideration to discuss the findings reported in the main manuscript. I warmly suggest to read some interesting works on nano clay-containing hydrogels: 1) https://doi.org/10.1021/acsestengg.2c00390 2) https://doi.org/10.1002/term.2115 3) https://doi.org/10.1038/s41467-019-11511-3
Comments on the Quality of English LanguageThe English is absolutely understandable.
Author Response
Dear Reviewer
Thank you very much for your time and preparing the review of this manuscript.
Please find the detailed responses below.
1. Mechanical Properties and Physical Intercalation: One notable limitation of the manuscript is the absence of information on the mechanical properties and physical intercalation of Cloisite® 30B in the polyurethane nanocomposite hydrogel patches. The authors should clarify whether Cloisite® 30B is intercalated or dispersed as macroscopic particles and investigate the influence of this phyllosilicate on the mechanical properties. A thorough viscoelastic analysis, coupled with TEM microscopy, would be beneficial to understand the material's behavior more comprehensively.
Mechanical and structural studies of polymer nanocomposites have been published in our previous works [1,2]. The type of nanocomposite was determined based on XRD analysis. All the nanocomposites samples containing 0.5 and 1% of Closite®30B nanofiller are exfoliated, but for systems with 1% of Clo, a diffraction peak of very low amplitude is observed and is related to the presence of the nanofiller in the form of aggregated stacks in the polymer matrix. We do not observe a shift this peak towards lower values of the 2theta angle, therefore, it does not form an intercalated structure. The mechanical properties of nanocomposites were investigated using DMA technique. We observed the improvement of mechanical properties by incorporation of clay and which can be controlled by varying the nanofiller content in these systems as well as molecular dynamics (measured by NMR technique) of the polymer chains in PU-PEO nanocomposites depending on montmorillonite content in polymeric materials.
2. Formal Adjustments: The manuscript should undergo formal adjustments regarding the expression of molecular weights of polymers. It is recommended to present the molecular weight in Daltons (Da) or without dimensions to ensure clarity and conformity to standard conventions.
The polymers used for the synthesis of nanocomposites were purchased from Sigma-Aldrich Co. In this work, we provided molecular weights in units consistent with the Sigma-Aldrich Co specifications, and we also wanted to be consistent with what we presented in our previous works.
3. Literature update: The literature panorama should be taken into consideration to discuss the findings reported in the main manuscript. I warmly suggest to read some interesting works on nano clay-containing hydrogels: 1) https://doi.org/10.1021/acsestengg.2c00390 2) https://doi.org/10.1002/term.2115 3) https://doi.org/10.1038/s41467-019-11511-3
Thank you very much for these literature suggestions, these publications are very interesting from an application point of view, we will use them in the next stages of work on research on the release of paracetamol directly into the skin. They will also be a valuable hint on how to interpret the results of, for example, mechanical measurements in systems in contact with tissues.
References:
1. Strankowska, J.; Piszczyk, Ł.; Strankowski, M.; Danowska, M.; Szutkowski, K.; Jurga, S.; Kwela, J. Molecular Dynamics Studies of Polyurethane Nanocomposite Hydrogels. Eur. Phys. J. Spec. Top. 2013, 222, 2179–2186, doi:10.1140/epjst/e2013-01994-8.
2. Miotke, M.; Strankowska, J.; Kwela, J.; Strankowski, M.; Piszczyk, Ł.; Józefowicz, M.; Gazda, M. Nanosize Effect of Clay Mineral Nanoparticles on the Drug Diffusion Processes in Polyurethane Nanocomposite Hydrogels. Eur. Phys. J. Plus 2017, 132, 401, doi:10.1140/epjp/i2017-11708-1.
Reviewer 2 Report
Comments and Suggestions for Authors
The article is not of sufficient quality to be published in this journal. Specific questions and points requiring attention are itemized below.
Reviewer comments: Line 37. What type of crosslinks? Please explain in detail.
Reviewer comments: Line 47. Why do the authors select Polyurethane as a polymer to produce hydrogels? Describe the reasons.
Reviewer comments: Line 48. What class of mechanical properties?
Reviewer comments: Line 50. Why are polyurethane materials widely used as wound dressing, adhesives, and elastomers in skin tissue engineering?? Explain in detail.
Reviewer comments: It is still difficult to find the novelty of the work concerning what has already been published. So, what is the difference between what is published and what the authors want to publish? It is not clear.
Reviewer Comments: The authors must evaluate the chemical identity by FTIR of the materials obtained.
Reviewer Comments: The authors must evaluate the porosity and internal microstructure of the materials obtained.
Reviewer Comments: Line 153: Why submerged in paracetamol/aqueous/ethanol solutions? Explain in detail.
Reviewer Comments: Where is the statistical analysis section????
Reviewer Comments: Line 252. In the swelling studies, the authors must use distilled water and PBS buffer as a control.
Reviewer Comments: Line 252. What pH values were used in this analysis?
Reviewers comments: The discussion sections is poor. More comparisons with previous literature should be discussed.
Reviewers comments: In tables, significant differences must be added.
Reviewers comments: The authors must use more references and more recent (2018-2023).
Author Response
Dear Reviewer
Thank you very much for your time and preparing the review of this manuscript.
Please find the detailed responses below.
Reviewer comments: Line 37. What type of crosslinks? Please explain in detail.
Two cross-linking agents, triethanolamine and glycerol, were used to synthesize PU/PEG hydrogels. The selected cross-linking agents have a specific structure and do not contain additives, that could affect the synthesis of the polyurethane systems. In addition, because of the relatively simple structure of cross-linking agents, we were able to adequately control the crosslinking process. As well as our laboratory experience with extenders containing hydroxyl groups (like butane-1,4-ol) and low molecule weight crosslinking compounds was the greatest. This is also particularly important in the context of appropriate substrate selection (proper determination of the ratio of NCO/OH groups) to obtain systems with the best possible thermal and mechanical properties.
Reviewer comments: Line 47. Why do the authors select Polyurethane as a polymer to produce hydrogels? Describe the reasons.
Currently, mainly acrylates, polyurethanes and silicones are used to produce patches for pharmaceutical use; unmodified silicones and acrylates have poor swelling properties, while polyurethanes can be chemically modified, e.g. cross-linked, to increase their sorption, which plays a key role in the active substance release processes. Considering PIB (polyisobutylene) plasters are being withdrawn from the market, polyurethanes are a good alternative to modern patches. In addition, our research team specializes in polyurethane materials, so this type of material is the basis of our scientific research.
Reviewer comments: Line 48. What class of mechanical properties?
The mechanical properties of polyurethane (PU) materials were studied in the context of dynamic-thermomechanical studies, where the storage modulus (E’), which is one of the indicators describing the performance of polyurethane nanocomposites, was analyzed. The developed systems, based on DMA studies showed a higher E' modulus for nanocomposites containing Cloisite® 30B, both in the glassy and viscoelastic region in comparison to the nonmodified PU. Mechanical, spectroscopic and structural studies of polymer nanocomposites have been published in our previous works [1–3]. The mechanical properties of nanocomposites were investigated using DMA technique. We observed the improvement of mechanical properties by incorporation of clay and which can be controlled by varying the nanofiller content in these systems as well as molecular dynamics (measured by NMR technique) of the polymer chains in PU-PEO nanocomposites depending on montmorillonite content in polymeric materials.
Reviewer comments: Line 50. Why are polyurethane materials widely used as wound dressing, adhesives, and elastomers in skin tissue engineering?? Explain in detail.
Polyurethane materials represent a wide range of polymers that can be easily modified by changing reactive substrates. Thanks to their good mechanical and thermal properties and characteristic viscoelastic properties, they can be successfully used as dressing materials. They can be some of the most biocompatible and safe for biological systems materials available today. Thanks to the ability to control and modify the structure of polyurethanes, it is possible to design systems with specific properties that may influence drug delivery processes, e.g. absorption, diffusion, solubility, erosion and degradation which makes it possible to use polyurethanes in a wide range of drug delivery systems [4].
Reviewer comments: It is still difficult to find the novelty of the work concerning what has already been published. So, what is the difference between what is published and what the authors want to publish? It is not clear.
The main aim of the work was to comprehensively analyze the effect of nanofiller (Cloisite® 30B) on the transport properties of the active substance in the proposed system. In the previous work, only one type of hydrogel was considered (PU/PEG 4000 TEA) for a 3% paracetamol concentration. In this work, we focus on the comparison of transport mechanism (swelling and release processes) for nanocomposite hydrogels, synthesized using different cross-linking agents (TEA, GLY), and we also answer the question of how different concentrations (1%, 5%) of paracetamol affect these processes. Thanks to this, it was possible to propose a new method of describing the transport process, taking into account the influence of cross-linking and the presence of nanoparticles on the efficiency of swelling and release of paracetamol from various nanocomposite matrices, especially the diffusion process description of short and long time approximation.
Reviewer Comments: The authors must evaluate the chemical identity by FTIR of the materials obtained.
Studies using FTIR and Raman spectroscopy are planned.
Reviewer Comments: The authors must evaluate the porosity and internal microstructure of the materials obtained.
The type of nanocomposite was determined based on XRD analysis. All the nanocomposites samples containing 0.5 and 1% of Closite®30B nanofiller are exfoliated, but for systems with 1% of Clo, a diffraction peak of very low amplitude is observed and is related to the presence of the nanofiller in the form of aggregated stacks in the polymer matrix. We do not observe a shift this peak towards lower values of the 2theta angle, therefore, it does not form an intercalated structure. The size of the pores in the PU/PEG 4000 hydrogels was examined using the DSC technique – thermoporometry and was presented in [2]. For the pure nanocomposite, the pore size was 4.27 nm, for 0.5% 4.57 nm, and for 1% Clo 6.84 nm.
Reviewer Comments: Line 153: Why submerged in paracetamol/aqueous/ethanol solutions? Explain in detail.
Paracetamol is very slightly soluble in water but dissolves well in ethanol (50 mg/mL). Ethanol is used as a drug delivery enhancer, this molecule enhances the permeation of both polar and nonpolar molecules into the skin, but the mechanism of this process is not fully explained [5].
Reviewer Comments: Where is the statistical analysis section????
We did not perform a typical statistical analysis because we did not compare the validity of using different models and choosing the best one. We assessed the correctness of using models based on the correlation coefficient r, and all measurement uncertainties were provided in the tables.
Reviewer Comments: Line 252. In the swelling studies, the authors must use distilled water and PBS buffer as a control.
The aim of the swelling studies was to introduce the active substance - paracetamol - into the hydrogel. For this purpose, paracetamol was dissolved in an aqueous ethanol solution. The solubility of paracetamol in PBS buffer is about 2 mg/mL, which is much less than ethanol.
Reviewer Comments: Line 252. What pH values were used in this analysis?
The pH tests during the measurements were not carried out because the polyurethane system showed an inert nature and the aqueous solutions were also inert. The relatively small addition of the active ingredient had little effect on the pH of the tested systems.
Reviewers comments: The discussion sections is poor. More comparisons with previous literature should be discussed.
The introduction section has been completed and comparisons to previous works have been added.
Reviewers comments: In tables, significant differences must be added.
Due to the fact that in the work we compare systems containing different concentrations of nanoparticles and other concentrations of paracetamol, pure systems with those containing Clo, and additionally, we also compare hydrogels with other cross-linking agents, it is difficult to mark the differences in the tables, which is why the discussion is carried out in the text of the work.
Reviewers comments: The authors must use more references and more recent (2018-2023).
Additional references to newer literature have been added to the corrected version of the manuscript.
References:
1. Strankowska, J.; Piszczyk, Ł.; Strankowski, M.; Danowska, M.; Szutkowski, K.; Jurga, S.; Kwela, J. Molecular Dynamics Studies of Polyurethane Nanocomposite Hydrogels. Eur. Phys. J. Spec. Top. 2013, 222, 2179–2186, doi:10.1140/epjst/e2013-01994-8.
2. Miotke, M.; Strankowska, J.; Kwela, J.; Strankowski, M.; Piszczyk, Ł.; Józefowicz, M.; Gazda, M. Nanosize Effect of Clay Mineral Nanoparticles on the Drug Diffusion Processes in Polyurethane Nanocomposite Hydrogels. Eur. Phys. J. Plus 2017, 132, 401, doi:10.1140/epjp/i2017-11708-1.
3. Miotke-Wasilczyk, M.; Józefowicz, M.; Strankowska, J.; Kwela, J. The Role of Hydrogen Bonding in Paracetamol–Solvent and Paracetamol–Hydrogel Matrix Interactions. Materials 2021, 14, 1842, doi:10.3390/ma14081842.
4. Wienen, D.; Gries, T.; Cooper, S.L.; Heath, D.E. An Overview of Polyurethane Biomaterials and Their Use in Drug Delivery. J. Controlled Release 2023, 363, 376–388, doi:10.1016/j.jconrel.2023.09.036.
5. Gupta, R.; Badhe, Y.; Rai, B.; Mitragotri, S. Molecular Mechanism of the Skin Permeation Enhancing Effect of Ethanol: A Molecular Dynamics Study. RSC Adv. 2020, 10, 12234–12248, doi:10.1039/D0RA01692F.
Reviewer 3 Report
Comments and Suggestions for Authors
In this study the Authors investigate the transport mechanism of paracetamol in polyurethane nanocomposites, specifically hydrogels containing Cloisite® 30B clay nanoparticles. The presence of nanoparticles affects swelling, showing a barrier effect, and their concentration influences aggregation and diffusion rates. The findings suggest practical applications, such as the controlled drug release in patches, by manipulating nanoparticle concentration and polymer matrix crosslinking. Overall, this research contributes to understanding drug transport in polyurethane nanocomposites, and can be published on Materials after minor revisions.
It would be interesting for the readers to add a characterization of the physical properties of the hydrogel (i.e. pore size, porosity..) to better understand the impact of these factors on swelling behavior and drug release kinetics.
Was the uniformity of paracetamol distribution measured?
Since the hydrogel is intended for biomedical applications, the understand of their behavior under physiological conditions could be interesting.
Please add error bars in all the graphs.
The English is mostly correct, but there are a few areas where clarity and readability can be improved.
Author Response
Dear Reviewer
Thank you very much for your time and preparing the review of this manuscript.
Please find the detailed responses below.
1. It would be interesting for the readers to add a characterization of the physical properties of the hydrogel (i.e. pore size, porosity..) to better understand the impact of these factors on swelling behavior and drug release kinetics.
Mechanical, spectroscopic and structural studies of polymer nanocomposites have been published in our previous works [1–3]. The type of nanocomposite was determined based on XRD analysis. All the nanocomposites samples containing 0.5 and 1% of Closite®30B nanofiller are exfoliated, but for systems with 1% of Clo, a diffraction peak of very low amplitude is observed and is related to the presence of the nanofiller in the form of aggregated stacks in the polymer matrix. We do not observe a shift this peak towards lower values of the 2theta angle, therefore, it does not form an intercalated structure. The mechanical properties of nanocomposites were investigated using DMA technique. We observed the improvement of mechanical properties by incorporation of clay and which can be controlled by varying the nanofiller content in these systems as well as molecular dynamics (measured by NMR technique) of the polymer chains in PU-PEO nanocomposites depending on montmorillonite content in polymeric materials.
The size of the pores in the PU/PEG 4000 hydrogels was examined using the DSC technique – thermoporometry and was presented in [2]. For the pure nanocomposite, the pore size was 4.27 nm, for 0.5% 4.57 nm, and 1% Clo 6.84 nm.
2. Was the uniformity of paracetamol distribution measured?
No, such studies have not been conducted.
3. Since the hydrogel is intended for biomedical applications, the understand of their behavior under physiological conditions could be interesting.
Yes, we agree with this and such studies are planned, especially in terms of the release of paracetamol into the skin under physiological conditions. The studies presented in this paper were aimed at understanding the mechanism of paracetamol release and the effectiveness of this process in the presence of nanoparticles in laboratory conditions to determine the appropriate release conditions into and through the skin.
4. Please add error bars in all the graphs.
Information about measurement errors has been added to the text of the manuscript. The uncertainty rectangles in the charts made them unreadable.
5. Comments on the Quality of English Language
The English is mostly correct, but there are a few areas where clarity and readability can be improved.
The English has been checked and all noticed errors have been corrected.
References:
1. Strankowska, J.; Piszczyk, Ł.; Strankowski, M.; Danowska, M.; Szutkowski, K.; Jurga, S.; Kwela, J. Molecular Dynamics Studies of Polyurethane Nanocomposite Hydrogels. Eur. Phys. J. Spec. Top. 2013, 222, 2179–2186, doi:10.1140/epjst/e2013-01994-8.
2. Miotke, M.; Strankowska, J.; Kwela, J.; Strankowski, M.; Piszczyk, Ł.; Józefowicz, M.; Gazda, M. Nanosize Effect of Clay Mineral Nanoparticles on the Drug Diffusion Processes in Polyurethane Nanocomposite Hydrogels. Eur. Phys. J. Plus 2017, 132, 401, doi:10.1140/epjp/i2017-11708-1.
3. Miotke-Wasilczyk, M.; Józefowicz, M.; Strankowska, J.; Kwela, J. The Role of Hydrogen Bonding in Paracetamol–Solvent and Paracetamol–Hydrogel Matrix Interactions. Materials 2021, 14, 1842, doi:10.3390/ma14081842.
Reviewer 4 Report
Comments and Suggestions for Authors
The manuscript presents the development of polyurethane/polyethylene glycol nanocomposite hydrogels containing Cloisite® 30B (PU/PEG/Cloisite® 30B), and the swelling and release mechanisms of the drug paracetamol in these hydrogels. The results are comprehensive and well-detailed, and the conclusions are to the point. Also, the authors have used relevant references and generally, the English language is in general correct.
I agree with its publication in Materials, after some necessary changes:
I think characterization results of the hybrid hydrogels are necessary for the complement of the article content. So, please provide characterization with techniques like NMR, FT-IR and TGA. For example, provide FT-IR spectra and TGA thermograms of (a) the pristine polyurethane, (b) clay (Cloisite 30B) and (c) polyurethane-clay nanocomposite.
Also, for the morphological analysis, provide SEM images of virgin polymer and with different loading of clay (PU/PEG/clay nanocomposite), to show the dispersity or agglomeration of the nanocomposite throughout the PU/PEG matrix, etc.
Author Response
Dear Reviewer
Thank you very much for your time and preparing the review of this manuscript.
Please find the detailed responses below.
I think characterization results of the hybrid hydrogels are necessary for the complement of the article content. So, please provide characterization with techniques like NMR, FT-IR and TGA. For example, provide FT-IR spectra and TGA thermograms of (a) the pristine polyurethane, (b) clay (Cloisite 30B) and (c) polyurethane-clay nanocomposite.
Also, for the morphological analysis, provide SEM images of virgin polymer and with different loading of clay (PU/PEG/clay nanocomposite), to show the dispersity or agglomeration of the nanocomposite throughout the PU/PEG matrix, etc.
Mechanical, spectroscopic and structural studies of polymer nanocomposites have been published in our previous works [1–3]. The type of nanocomposite was determined based on XRD analysis. All the nanocomposites samples containing 0.5 and 1% of Closite®30B nanofiller are exfoliated, but for systems with 1% of Clo, a diffraction peak of very low amplitude is observed and is related to the presence of the nanofiller in the form of aggregated stacks in the polymer matrix. We do not observe a shift this peak towards lower values of the 2theta angle, therefore, it does not form an intercalated structure. The mechanical properties of nanocomposites were investigated using DMA technique. We observed the improvement of mechanical properties by incorporation of clay and which can be controlled by varying the nanofiller content in these systems as well as molecular dynamics (measured by NMR technique) of the polymer chains in PU-PEO nanocomposites depending on montmorillonite content in polymeric materials.
The size of the pores in the PU/PEG 4000 hydrogels was examined using the DSC technique – thermoporometry and was presented in [2]. For the pure nanocomposite, the pore size was 4.27 nm, for 0.5% 4.57 nm, and 1% Clo 6.84 nm.
References:
1. Strankowska, J.; Piszczyk, Ł.; Strankowski, M.; Danowska, M.; Szutkowski, K.; Jurga, S.; Kwela, J. Molecular Dynamics Studies of Polyurethane Nanocomposite Hydrogels. Eur. Phys. J. Spec. Top. 2013, 222, 2179–2186, doi:10.1140/epjst/e2013-01994-8.
2. Miotke, M.; Strankowska, J.; Kwela, J.; Strankowski, M.; Piszczyk, Ł.; Józefowicz, M.; Gazda, M. Nanosize Effect of Clay Mineral Nanoparticles on the Drug Diffusion Processes in Polyurethane Nanocomposite Hydrogels. Eur. Phys. J. Plus 2017, 132, 401, doi:10.1140/epjp/i2017-11708-1.
3. Miotke-Wasilczyk, M.; Józefowicz, M.; Strankowska, J.; Kwela, J. The Role of Hydrogen Bonding in Paracetamol–Solvent and Paracetamol–Hydrogel Matrix Interactions. Materials 2021, 14, 1842, doi:10.3390/ma14081842.
Round 2
Reviewer 1 Report
Comments and Suggestions for Authors
The authors have somehow considered the Reviewers' concern and thus the manuscript is suitable for publication.
Comments on the Quality of English Languagesome sentence is too complex and can be rephrased to be much more linear. However, the average quality is fine.
Reviewer 2 Report
Comments and Suggestions for Authors
The article can be accepted
Reviewer 4 Report
Comments and Suggestions for Authors
I do not have any further comments. It can be published in the journal.